# The role of the tropical carbon balance in determining the large atmospheric CO<sub>2</sub> growth rate in 2023

Liang Feng<sup>1,2</sup>, Paul I. Palmer<sup>1,2</sup>, Luke Smallman<sup>1,2</sup>, Jingfeng Xiao<sup>3</sup>, Paolo Cristofanelli<sup>4</sup>, Ove Hermansen<sup>5</sup>, John Lee<sup>6</sup>, Casper Labuschagne<sup>7</sup>, Simonetta Montaguti<sup>4</sup>, Steffen M. Noe<sup>8</sup>, Stephen M. Platt<sup>5</sup>, Xinrong Ren<sup>9</sup>, Martin Steinbacher<sup>10</sup>, Irène Xueref-Remy<sup>11</sup>

1: University of Edinburgh, Edinburgh, EH9 3FF, UK.

20

- 2: National Centre for Earth Observation, University of Edinburgh, Edinburgh, EH9 3FF, UK.
- 10 3: Earth Systems Research Center, University of New Hampshire, Durham, NH 03824, USA.
  - 4: National Research Council of Italy Institute for Atmospheric Sciences and Climate, I-40129 Bologna, Italy.
  - 5: Norwegian Institute for Air Research, Instituttveien 18, 2007 Kieller, Norway.
  - 6: School of Forestry, CRSF, University of Maine, ME, 04469, USA.
  - 7: South African Weather Service, Cape Point research station, Stellenbosch 7599, South Africa.
- 5 8: Institute of Forestry and Engineering, Estonian University of Life Sciences, 51014 Tartu, Estonia.
  - 9: NOAA Air Resources Laboratory Atmospheric Sciences and Modeling Division, College Park, MD 20740, USA.
  - 10: Laboratory for Air Pollution / Environmental Technology, Swiss Federal Laboratories for Materials Science and Technology (Empa), 8600 Duebendorf, Switzerland.
  - 11: Institut Méditerranéen de Biodiversité et d'Ecologie marine et continentale, Aix-Marseille Univ, Avignon Univ, CNRS, IRD, 13397 Marseille, France.

Correspondence to: Liang Feng (liang.feng@ed.ac.uk) and Paul Palmer (pip@ed.ac.uk)

Abstract. The global annual mean atmospheric CO<sub>2</sub> growth rate in 2023 was one of the highest since records began in 1958, comparable to values recorded during previous major El Niño events. We do not fully understand this anomalous growth rate, although a recent study highlighted a role for boreal North American forest fires. We use a Bayesian inverse method to interpret global-scale atmospheric CO<sub>2</sub> data from the US National Aeronautics and Space Administration (NASA) Orbiting Carbon Observatory (OCO-2). The resulting *a posteriori* CO<sub>2</sub> flux estimates reveal that from 2022 to 2023 the biggest changes in CO<sub>2</sub> fluxes of net biosphere exchange (NBE) – for which positive values denote a flux to the atmosphere – were over the land tropics. We find that the largest NBE increase is over eastern Brazil, with small increases over southern Africa and Southeast Asia. We also find significant increases over southeast Australia, Alaska, and western Russia. A large NBE increase over boreal North America, due to fires, is driven by our *a priori* inventory, informed by independent data. The largest NBE reductions are over western Europe, USA, and central Canada. Our NBE estimates are consistent with gross primary production estimates inferred from satellite observations of solar induced fluorescence and with satellite observations of vegetation greenness. We find that warmer temperatures in 2023 explain most of the NBE change over eastern Brazil, with hydrological changes more important elsewhere across the tropics. Our results suggest that ongoing environmental degradation of the Amazon is now playing a substantial role in increasing the global atmospheric CO<sub>2</sub> growth rate.

## 1 Introduction

45

65

The annual mean growth rate of atmospheric carbon dioxide (CO<sub>2</sub>) is widely used as a zeroth order metric to determine the health of our planet. Even from the first few years' worth of data collected at Mauna Loa in the late 1950s it was plain to see that a) land vegetation imposed a large seasonal cycle on atmospheric CO<sub>2</sub> via photosynthesis and respiration, and b) combustion of fossil fuels led to a planetary scale impact on the atmosphere (Keeling, 1960; Keeling et al., 1976). Changes in the annual accumulation of atmospheric CO<sub>2</sub> (growth rate), the magnitude and phase of the seasonal cycle, and how they vary geographically, provide important clues about economic activity and the health of the land biosphere (Keeling et al., 1996; Graven et al., 2013; Barlow et al., 2015). These changes are inextricably linked, e.g., elevated uptake by the land biosphere will influence the annual growth rate as well as the seasonal cycle (e.g., Ainsworth and Rogers, 2007). On a global scale, using mass balance arguments, we know that only about 44% of fossil fuel emissions of CO<sub>2</sub> remain in the atmosphere (the airborne fraction) (Bennett et al., 2024) with the land biosphere and oceans absorbing the other 56%, approximately equally but with substantial year to year changes (Friedlingstein et al., 2023). The quasi-stability of the airborne fraction suggests that the land biosphere and the oceans absorb a progressively larger absolute amount of CO<sub>2</sub> from the atmosphere. We have an incomplete understanding of where this carbon is being absorbed and the stability of the resulting accumulated terrestrial carbon reservoirs against future changes in climate, e.g. Armstrong McKay et al., (2022). Consequently, years in which there are anomalously large annual mean CO<sub>2</sub> growth rates prompt concern from the scientific community. This concern grows when state-of-the-art process-based land biosphere models cannot forecast or explain these anomalies (Kondo et al., 2020).

Figure 1 shows the annual mean CO<sub>2</sub> growth rates reported by NOAA on a global scale, determined by combining data collected at sites across the globe, and from Mauna Loa in Hawaii (19.5°N, 155.6°W), USA, a site typically assumed to be representative of changes in the northern hemisphere carbon cycle (Buermann et al., 2007). The global picture shows that 2023 (Figure 1a) had one of the largest CO<sub>2</sub> growth rates on record, typically associated with the El Niño phase of ENSO, e.g., 1986, 1997/1998, and 2015/2016. What is also evident is a progressive increase in the annual growth rates from the 1950s (Figure 1c). Even anomalous values recorded in the last quarter of the 20<sup>th</sup> century are close to the median value from the 21<sup>st</sup> century (Figure 1c). The corresponding data collected at Mauna Loa shows a slightly different picture for the annual CO<sub>2</sub> growth rate (Figure 1b). At this site, the growth rate in 2023 was the largest on record, exceeding the past peak growth during 1997/1998 El Niño, attributed to extensive burning of peat over Southeast Asia (Page et al., 2002), and the 2015/2016 El Niño (Liu et al., 2017). At Mauna Loa, progressive changes in the growth rates are slightly more exaggerated than global mean values (Figure 1b,d), suggesting a larger role for tropical latitudes.

Data-driven top-down flux inversions allow us to attribute these observed changes in the atmospheric CO<sub>2</sub> growth rate to regional changes in surface carbon fluxes. Estimating regional carbon fluxes from atmospheric data requires an atmospheric transport model that describes the physical relationship between surface CO<sub>2</sub> fluxes and the resulting atmospheric distribution

of CO<sub>2</sub>, *a priori* estimates of the distribution and magnitude of fluxes, and a Bayesian inference method that fits this model to the data accounting for model and data uncertainties (Tans et al., 1990; Baker et al., 2006; Gurney et al., 2002, 2004). Using an atmospheric transport model introduces additional errors (Schuh et al., 2019; Oda et al., 2023) but it remains an essential tool for interpreting the atmospheric data. Satellite observations of atmospheric CO<sub>2</sub> have challenged current understanding of the carbon cycle (Liu et al., 2017; Chatterjee et al., 2017; Patra et al., 2017; Palmer et al., 2019; Wang et al., 2020; Basso et al., 2023; Hugelius et al., 2024; O'Sullivan et al., 2024; Liu et al., 2024). They have primarily achieved this by collecting data over geographical regions that are not well covered by ground-based networks, particularly over the land tropics. These datasets are typically available with a time lag of only a few months, enabling us to explain the reasons behind anomalous annual CO<sub>2</sub> growth rates within a year of them happening.

To interpret recent annual changes in the CO<sub>2</sub> growth rate, we use the global 3-D GEOS-Chem atmospheric transport model and an ensemble Kalman filter to adjust our *a priori* distribution of CO<sub>2</sub> flux estimates to fit *in situ* and satellite observations of atmospheric CO<sub>2</sub>. These methods and data are described in the next section. We report our results in Sect 3 and conclude our study in section 4.

## 2 Data and Methods

Here, we describe the modelling framework we use to infer *a posteriori* spatial distributions of CO<sub>2</sub> fluxes, 2014—2023, from atmospheric data and *a priori* inventories flux estimates, and the auxiliary atmospheric and land surface we use to evaluate the resulting *a posteriori* flux estimates.

#### 2.1 Inversion Framework

We use the GEOS-Chem global 3-D atmospheric chemistry transport model of version 13.4 to provide the relationship between the surface fluxes and changes in atmospheric CO<sub>2</sub>. For the experiments we report, we run the model at a horizontal resolution of 2° (latitude) × 2.5° (longitude), driven by Modern-Era Retrospective Analysis for Research and Applications, version 2 (MERRA2) meteorological reanalyses from the Global Modeling and Assimilation Office (GMAO) based at NASA Goddard Space Flight Center (GSFC).

We use *a priori* CO<sub>2</sub> flux inventories, which include year-specific monthly biomass burning emission (GFEDv4.1; Randerson et al., 2017), and year-specific monthly anthropogenic emissions (ODIAC; Oda et al., 2018; Oda and Maksyutov, 2021). The anthropogenic emission estimates were extended to 2023 under the assumption that these emissions from the southern hemisphere remain stable between 2022 and 2023 but increased by 1.4% over the northern hemisphere based on data reported in the 2024 Statistical Review of World Energy by the Energy Institute. We use year-specific terrestrial biosphere fluxes with a temporal resolution of three hours (CASA; Olsen and Randerson, 2004) up to the end of 2018, and repeat values for 2018 in

subsequent years. We use monthly climatological ocean fluxes (Takahashi et al., 2009), which we scale uniformly to a global annual uptake of 2.5 PgC yr<sup>-1</sup>, 2014-2024, inclusively, following Nassar et al. (2010).

We use an established EnKF framework to estimate surface CO<sub>2</sub> fluxes, 2014—2023, inclusively, from atmospheric CO<sub>2</sub> data collected by OCO-2 and the US National Oceanic and Atmospheric Administration (NOAA) *in situ* ground-based observation network, 2014—2023, inclusively. For brevity, we provide a summary of the approach and refer the reader to other papers for further details (Feng et al., 2009, 2017; Palmer et al., 2019).

Adopting a widely used approach, we assume that the fossil fuel emissions are well known and estimate monthly *a posteriori* natural CO<sub>2</sub> fluxes, including fire emissions, terrestrial and ocean biospheric CO<sub>2</sub> fluxes, which are approximated by (Feng et al., 2017):

$$f_a(x,t) = f_o(x,t) + \sum_i c_i BF_i(x,t),$$
 (1)

where  $f_a(x,t)$  and  $f_0(x,t)$  describes the *a posteriori* and *a priori* CO<sub>2</sub> flux estimate at location x and time t, respectively. The pulse-like basis functions  $BF_i(x,t)$  represent the sum of natural fluxes used to represent their overall spatial pattern over each pre-defined sub-region. The coefficients  $c_i$  form the state vector to be estimated by optimally fitting the model to the data.

We define our land sub-regions by further dividing each of the 11 TransCom-3 land regions (Gurney et al., 2002) into 30 nearly equal sub-regions, with the exception for temperate Eurasia that has been divided into 56 sub-regions, due to its large landmass. We divide the 11 TransCom-3 ocean regions into 132 sub-regions. Our state vector includes monthly scaling factors for 488 regional pulse-like basis functions that describe natural CO<sub>2</sub> fluxes, including 356 land regions and 132 oceanic regions (Figure A1). We determine these coefficients by optimally fitting the corresponding atmospheric model concentrations with *in situ* and OCO-2 data (Feng et al., 2017):

$$\boldsymbol{c}_a = \boldsymbol{c}_f + \mathbf{K}[\mathbf{y} - H(\boldsymbol{c}_f)], \tag{2}$$

where  $c_a$  and  $c_f$  denote the *a posteriori* and *a priori* state vectors, respectively,  $\mathbf{y}$  denotes satellite and *in situ* CO<sub>2</sub> observations, and H describes the observation operator that relates surface fluxes (i.e., the coefficients) to the observations. Here we sample the 3-D GEOS-Chem model CO<sub>2</sub> fields at the time and location of each observation. For comparison with OCO-2 XCO2 retrievals, we further convolve the resulting model profiles with scene-dependent OCO-2 averaging kernels. In our EnKF framework, we introduce a flux perturbation (coefficients) ensemble  $\Delta \mathbf{C}$  to represent the *a priori* error covariance, and calculate the Kalman gain matrix  $\mathbf{K}$  in Eq. (2) by using

$$\mathbf{K} = \Delta \mathbf{C} \Delta \mathbf{Y}^{\mathrm{T}} [\Delta \mathbf{Y} \Delta \mathbf{Y}^{\mathrm{T}} + \mathbf{R}^{-1}]^{-1}, \tag{3}$$

where **R** is the observation error covariance, and  $\Delta Y = H(\Delta C)$  represents the projection of the flux perturbation ensemble to observation space, which is based on the same GEOS-Chem model run at the same horizontal resolution of 2° (latitude) × 2.5°

(longitude) as our *a priori* simulations. We use a four-month moving lag window to reduce the computational costs for projecting the flux perturbation ensemble into observation space long after their emissions (in this case longer than four months), beyond which time it is difficult to distinguish between the emitted signal from variations in the ambient background atmosphere (Feng et al., 2016). To calculate sequentially the *a posteriori* estimate and the associate uncertainty via Eqs (2) and (3) we use an efficient numerical LU solver (Feng et al., 2017).

For simplicity we assume a fixed uncertainty of 40% for coefficients corresponding to *a priori* CO<sub>2</sub> fluxes over each sub-140 region. We assume that *a priori* errors are correlated with a spatial correlation length of 500 km over land, and 800 km over oceans, and with a temporal correlation of one month. Our experiments show that our results, such as the estimated changes in *a posteriori* CO<sub>2</sub> fluxes between 2022 and 2023 and between 2022 and 2024, are largely insensitive to differences assumptions about *a priori* uncertainties (±10%) and correlation length scales (±100 km).

#### 145 2.2 In situ and OCO-2 atmospheric CO<sub>2</sub> data

150

We use version v11r of OCO-2 retrievals of column average dry air mole fraction (XCO2) from the NASA's Jet Propulsion Laboratory (JPL) Atmospheric CO<sub>2</sub> Observation from Space (ACOS) team (Taylor et al., 2023). We only assimilate the nadir and glint observations over land, considering possible bias between the land and ocean XCO2 data. The consequent poor observational coverage over the ocean could result in the disaggregation of the land and ocean CO<sub>2</sub> fluxes being more sensitive to the *a priori* ocean flux inventory. Through sensitivity studies we find that our land CO<sub>2</sub> flux anomalies are not significantly sensitive to the to the *a priori* ocean flux inventory (not shown) or to the absence of OCO-2 glint data (Figure A2). To reduce the computational costs and error correlations, we thinned the OCO-2 observations to ensure a minimal time interval of 10 s.

We also assimilate *in situ* measurements of CO<sub>2</sub> mole fraction data from a subset of 113 sites (Figure A1) included in the NOAA GLOBALVIEWPlus 8.0 data product (Schuldt et al., 2022), incorporating data from the Integrated Carbon Observation System (ICOS RI et al., 2024).

# 2.3 GOSIF Gross Primary Productivity (GPP)

We use a global GPP product that is based on OCO-2 solar induced fluorescence (GOSIF) and linear relationships between solar induced fluorescence (SIF) and GPP (Li and Xiao, 2019). We chose this data product, available globally at a spatial resolution of  $0.05^{\circ}$  and a temporal resolution of eight days, because it is close to the median of observation-derived GPP estimates (Li and Xiao, 2019) and is available over our study period. The mean annual global total for CO<sub>2</sub> (2000-2023) is  $135.5 \pm 8.8 \text{ Pg C yr}^{-1}$ , with a significant upward trend over the northern hemisphere. Comparisons show that this GPP data product is highly correlated (R<sup>2</sup>=0.74) with GPP measurements collected at 91 eddy covariance flux sites across the globe.

Here, we use the monthly mean dataset and re-grid it to a regular one-degree grid to compare it with other variables including our *a posteriori* CO<sub>2</sub> flux estimates.

# 2.4 Gravity Recovery And Climate Experiment (GRACE) data

The GRACE space mission was jointly developed by NASA and DLR (German Space Agency) and launched into space in 2002. It measures temporal variations of the Earth's gravity field by tracking, using a K-band ranging system, the inter-satellite range and range rate between two coplanar, low altitude satellites (Tapley et al., 2004). The GRACE Science Data System uses these measurements, along with ancillary data, to estimate monthly (or sub-monthly) time series of global Earth's gravity fields (Bettadpur, 2007; Flechtner, 2007). Here, we use the NASA GRCTellus GRACE land product (RL06.2) for monthly total water storage (liquid water equivalent depth) at  $1^{\circ} \times 1^{\circ}$  global grids from January 2014 through March 2024 (http://grace.jpl.nasa.gov/). We have used these data in our previous studies, e.g., Feng et al., (2022, 2023).

# 2.5 NASA meteorological reanalyses

We use surface temperature (T<sub>S</sub>), specific humidity (SH), soil moisture in the top 0—10 cm (ground wetness, WET) datasets from MERRA2 developed by the GMAO at NASA GSFC to study environmental changes from 2010 to 2023. We calculate the vapour pressure deficit (VPD) from the 10-m MERRA2 temperature, and specific humidity following Fang et al. (2022). We have used these reanalyses data previously to study *a posteriori* CO<sub>2</sub> fluxes (Palmer et al., 2019) and methane emissions (Feng et al., 2022, 2023).

In Appendix B, to examine the robustness of the results reported from our control run, described above, we report results from three sensitivity inversion that use different meteorological reanalyses, *a priori* inventories, and additional ocean sun-glint data collected by OCO-2. These sensitivity calculations provide confidence that the result we report in this study is robust.

## 3 Results

Figure 2 shows *a posteriori* net fluxes of CO<sub>2</sub> on a global scale, and across southern, tropical, and northern latitudes to provide some broad geographical context. These values are broadly consistent with annual values for the atmospheric CO<sub>2</sub> growth rates – an important zeroth order assessment of our *a posteriori* net fluxes. Our value for 2023 inferred from OCO-2 data is 3.0 ppm yr<sup>-1</sup>, about 0.2 ppm yr<sup>-1</sup>higher than the value inferred from NOAA CO<sub>2</sub> mole fraction data. We acknowledge that CO<sub>2</sub> growth rate estimates inferred from NOAA data can depart from the true value based on whole-atmosphere CO<sub>2</sub> changes (Pandey et al., 2024). Building on ongoing our model evaluation, e.g., Deng et al., (2024) and Friedlingstein et al., (2024), we find that the *a posteriori* CO<sub>2</sub> concentrations for 2023 are generally within 0.5 ppm of data collected by spectrometers from the Total Carbon Column Observing Network (TCCON) (Wunch et al., 2011), with a standard deviation smaller than 1.2 ppm.

As expected, the largest contribution of the global net flux originates from the northern hemisphere (Figure 2d), where there is a superposition of boreal and midlatitude ecosystems that contribute to the global uptake of CO<sub>2</sub> and large cities and other emission hotspots. At these latitudes, the year-to-year variations are comparatively small, limited to << 1PgC, and in the last two years since the 2021 peak there has been a small decrease in net emissions to pre-pandemic values (3.38—3.96 PgC yr<sup>-1</sup>, 2014—2020). Over our study, these changes have typically represented 62—92% of the global budget, with the smallest values typically during El Niño years when the tropics plays a larger role. The tropics show large year-to-year changes over our study period (Figure 2c) with a large peak in emissions that we have not observed since the 2015/2016 El Niño. We find the large increase in net CO<sub>2</sub> fluxes predominately originates from the tropics, representing 21% in 2022 and 38% in 2023. Our calculations suggest that this anomalous increase in tropical CO<sub>2</sub> flux in 2023 is explained mainly by an increased CO<sub>2</sub> flux over East Amazon (Figure A3). The net uptake in the southern hemisphere (Figure 2b) also shows a similar but small year-to-year change with the highest uptake in the last years, consequently compensating for emissions elsewhere on the globe. The 16% decrease in net uptake in 2023 reduced the influence of this region on the global net flux, reinforcing the role of the tropics on the global scale.

Figure 3 shows annual spatial distributions of the annual change in the net biosphere exchange (NBE) – the net CO<sub>2</sub> flux minus the a priori fossil fuel emissions removed – from 2022 to 2023 and as a comparison from 2014 to 2015 when there was a comparably largest change in the growth rate associated with the 2015/2016 El Niño. This widely used subtraction approach to determine NBE implicitly assumes perfect knowledge of fossil fuel combustion of CO<sub>2</sub>, but we acknowledge that making that assumption has implications for NBE estimates, although this is minimal over the tropics where anthropogenic emissions are comparatively small (Oda et al., 2023). A positive annual change in NBE represents a larger net amount of CO<sub>2</sub> to the atmosphere. We find that the largest positive increases in NBE are found across the tropics, with peak values over eastern Brazil, southern Africa, eastern and southern China, mainland and maritime Southeast Asia, and Southeast Australia. The emission hotspot over western Canada is from wildfires (Byrne et al., 2024) but our a posteriori feature is almost exclusively from the a priori inventory, determined by independent satellite data, because large aerosol optical depths over and downwind of these extensive fires where OCO-2 data are unreliable; Byrne et al. (2024) inferred carbon emissions from these fires using satellite observations of carbon monoxide. We also find large positive increases in NBE over Alaska and Russia. Regions with elevated uptake in 2023 are limited to the US and central Canada, mainland Europe, with weaker uptake over Siberia, Turkey, and some parts of East Africa. In comparison, the tropics in 2015 shows regions with positive and negative changes in NBE over tropical South America, a large increase over East and Central Africa (Palmer et al., 2019), with some of the largest increases over mainland and maritime Southeast Asia, as we also found in 2023. Elevated uptake was mainly confined to boreal latitudes. These changes in a posteriori fluxes are broadly consistent with independent estimates of GPP changes inferred from the OCO-2 SIF data product and from vegetation greenness, providing us with some confidence that our estimated fluxes are physically plausible. The annual mean budgets for individual geographical regions where we see the largest changes in NBE (rectangles in Figure 3a), show that East Amazon is almost exclusively responsible for the large increase in pan-tropical CO<sub>2</sub> flux in 2023, with a smaller contribution from Southeast Asia.

Figure 4 shows the geographical distribution of changes in parameters that describe large-scale CO<sub>2</sub> flux changes – temperature and water availability. Geographical locations where we report the largest increases in NBE (and largest reductions in GPP) in 2023, e.g., Brazil, southern Africa, southeast Australia, are coincident with locations where we saw some of the largest increases in temperature, VPD, and the largest reductions in LWE. Where we reported the largest decreases in NBE (and largest increases in GPP), e.g. parts of the contiguous US and central Canada, we saw cooler temperatures and lower VPDs, and small increases in LWE. We find a similar level of consistency between the data products and meteorological reanalyses in 2015. Recent work using an ensemble of dynamic global vegetation models highlighted the detrimental impact of warming on tropical ecosystems (Sitch et al., 2024), consistent with our results.

Figure 5 describes these relationships more quantitatively by using linear and quadratic multivariate fits of MERRA2 rainfall. temperature, and soil moisture anomalies to our a posteriori NBE anomalies, 2014—2023, inclusively, over the geographical regions highlighted in Figure 3a. For the linear fits (f1), we assume that the a posteriori NBE anomalies are a linear function of MERRA2 rainfall (R), surface temperature (Ts), and soil moisture (SM) anomalies:  $\triangle NBE = \Delta_0 + \alpha_R \Delta R + \alpha_T \Delta T_S + \alpha_{SM} \Delta SM$ , where  $\Delta$  denotes an anomaly,  $\alpha_x$  denotes the regression coefficient for a particular variable x,  $\Delta_0$  denotes the fitting residual. We scale these anomalies by their respective standard deviations and smooth them by applying a four-month moving window to reduce the noises and (partially) account for the time lag between flux and environmental drivers. We use a least-square method to estimate the four regression coefficients, which we report in Table A1, with results from our sensitivity tests shown in Table B2. We also consider a quadratic regression model (f2) to explain NBE anomalies, including linear and quadratic terms for the same three quantities used in the linear model but without cross terms, and found this only marginally outperforms the linear model. Both models are statistically significant, with p values < 0.001, so for simplicity of interpretation we use the linear fits. In sensitivity calculations, we find that changes in VPD or LWE do not improve the fits to NBE anomalies. The models capture most of the NBE changes, with the notable exception of mid 2022 when our NBE fluxes shows a sharp increase that is not explained by temperature or water. Based on the normalized linear fitting coefficients, we find for these fits that changes in temperature explain most of the NBE changes we observe over East Amazon (Table A1 and Figure B3), but soil moisture changes are more important over Northern tropical Africa, southern Africa, and tropical Asia (Table A1 and Figure B3). Rainfall changes are more important over Southeast Asia. Independent GOSIF GPP estimates determined from satellite SIF observations (Li and Xiao, 2019) show a significant decrease from 2022 to 2023 over tropical regions, particularly over eastern Amazonia, southern Africa, tropical Asia and Southeast Asia (Figure A4), consistent with the increase we report for our a posteriori NBE estimates (Figure 5). More generally, we find that changes in GOSIF GPP are better than other individual predictors at describing our a posteriori CO<sub>2</sub> flux anomalies over Tropical Asia, Southeast Asia, and southern Africa. Table A2 shows the permutation importance of individual predictors in our multivariate linear models.

# **4 Concluding Remarks**

290

We reported regional changes in the net biospheric exchange (NBE) of CO<sub>2</sub> inferred from OCO-2 retrievals of XCO2 from 2022 and 2023 to examine the origin of the large atmospheric growth rate reported for that period. Positive values of NBE denote net CO<sub>2</sub> fluxes to the atmosphere. We find that most of the increase in atmospheric CO<sub>2</sub> in 2023 is due to increased NBE over the land tropics, supported by a modest reduction in uptake in southern extratropics, in agreement with a recent study (Gui et al., 2024). Further examination of our results revealed increased NBE over eastern Brazil, southern Africa. eastern and southern China, mainland and maritime Southeast Asia, and Southeast Australia. Extensive wildfires over western Canada during boreal summer months also substantially contributed to the atmospheric CO<sub>2</sub> growth rate in 2023 (Byrne et al., 2024), but in terms of atmospheric CO<sub>2</sub> this information is exclusively from the a priori inventory that is determined by independent satellite data. We also find increased uptake (lower NBE values) over the US and central Canada, mainland Europe, with weaker uptake over Siberia, Turkey, and some parts of East Africa. These large-scale patterns of NBE are consistent with data-driven estimates of gross primary production and vegetation greenness, and with changes in surface temperature, rainfall, and surface water (Figures 4 and B3). We find that warmer temperatures in 2023 explain most of the change in NBE over eastern Brazil, with changes in hydrological quantities - rainfall or soil moisture - more important elsewhere across the tropics. Additional knowledge is needed to help reconcile CO2 flux estimates from land biosphere processbased models and those inferred from inversions (Kondo et al., 2020). Our quantitative exploration of the relationships between our a posteriori NBE anomalies and changes in environmental parameters (Figure 5) helps to interpret observed changes in atmospheric CO<sub>2</sub> but can also help to evaluate and improve process-based land biosphere models.

Our main analysis has focused on 2023, but it is important to put this one year into a broader historical context, at least in the past decade when we have seen a marked increase in atmospheric growth rates of atmospheric CO<sub>2</sub> (Figure 1). Some of this increase can be explained by changes in fossil fuel combustion and other forms of human activity, but the largest spikes in atmospheric CO<sub>2</sub> growth rates coincide with years when there is a strong El Niño event (Figure 1), primarily associated with large-scale perturbations to the hydrological cycle that impact tropical ecosystems. In strong El Niño years, such as 2015/2016, widespread droughts reported across the tropics (Jiménez-Muñoz et al., 2016) resulted in a notable increase in fires (Liu et al., 2017) and can in some ecosystems lead to a widespread loss of tree density and a change of the floristic composition (Prestes et al., 2024).

In 2023, the multivariate El Niño Southern Oscillation index, indicative of El Niño and La Niña strength, was approximately half the value of recent El Niño events, such as 2015/2016. There are distinct differences in the spatial patterns of rainfall, atmospheric aridity (given by vapour pressure deficit), and soil moisture over the tropics (Figure 4). But the loss of carbon sequestration in 2023 and 2015/2016 was comparable. Our findings highlight the complex response of the tropical biosphere to environmental change, reflecting differences in the sensitivity and vulnerability of plants to localized droughts and

increasing surface temperature (Table A1). Further quantifying these different sensitivities using independent *in situ* ecological observations will significantly improve our ability to model important biospheric processes in terms of atmospheric-biosphere carbon exchange, e.g., Liu et al. (2024).

We have extended our analysis to 2024, which is reported in Appendix C. We find that the reduced carbon uptake continues into 2024. Uptake by the Amazon basin in 2024 remains weaker than in 2022. There is also weakened uptake over southern tropical Africa (south of 20°S) and over tropical Asia. There is a small increase in uptake over temperate North America in 2024 compared to 2023. The resulting global net emission estimate for 2024 is 6.84±0.80 PgC, corresponding to a global CO<sub>2</sub> growth rate of 3.28±0.30 ppm yr<sup>-1</sup>.

Our interpretation of the OCO-2 column data suggests that the reduced uptake of CO<sub>2</sub> from tropical ecosystems played a key role in determining the anomalously large atmospheric CO<sub>2</sub> growth rates in 2023 and in 2024 (Appendix C). Our work is largely consistent with a recent independent study (Gui et al., 2024) that used the same OCO-2 data, but interpreted them with an independent atmospheric transport model, driven by different fossil fuel inventories and by AI-based dynamic global vegetation models. They also used a different inverse method approach. However, our results and those reported by Gui et al., (2024) are inconsistent with another independent study (Ke et al., 2024), based on a set of land biosphere models and an inversion experiment from the Copernicus Atmosphere Monitoring Service (CAMS). They significantly differ in the spatial patterns of carbon release and uptake. Resolving these discrepancies is beyond the scope of this work, but ultimately they do need to be resolved if we are to use these models to predict how global ecosystems will respond to a warming climate and an accelerated hydrological cycle, and the subsequent impacts on the carbon cycle (Armstrong McKay et al., 2022). If our main result is accurate – a moderate El Niño event, in the context of exceptional drought attributed to climate change (Clarke et al., 2024), has led to a significant reduction in carbon uptake by the tropical land biosphere – we might be observing the beginning of a decline in the ability of tropical ecosystems to absorb carbon. The long-term nature of this situation is unclear without further data, although the preliminary estimate of the 2024 atmospheric CO2 growth rate of 3.75±0.08 ppm vr<sup>-1</sup> is unprecedented since these records began in the late 1950s (https://gml.noaa.gov/ccgg/trends/gl gr.html; last access: 15th April 2025). A coordinated measurement campaign is urgently needed to document how tropical ecosystems are changing, whether these changes compromise the future ability to absorb and store carbon, and whether prolonged drought will substantially delay any ecosystem recovery.

Regularly reporting regional CO<sub>2</sub> fluxes with minimal delay, and interpreting them using auxiliary data, e.g., related to fire (such as the extensive North American boreal forest fires in 2023) and hydrology, are enabled by massive-scale international investment in satellite instruments that complement the detailed information provided by ground-based measurement networks. Collectively, these efforts provide vast volumes of information about the state of the planet at a time when we are observing unprecedented environmental changes. These data and the analysis tools needed to infer CO<sub>2</sub> fluxes collectively represent an

invaluable scientific resource that must be used to deliver frequent actionable information for policy makers. The agreement and divergence between our results and those from other independent studies underscore the efficacy and the shortcomings of the prevailing frameworks.

### **Code Availability**

The community-led GEOS-Chem model of atmospheric chemistry and transport model is maintained centrally by Harvard University (<a href="https://geoschem.github.io/">https://geoschem.github.io/</a>, last access: 5 May 2025), and is available on request. The ensemble Kalman filter code is publicly available as PyOSSE (<a href="https://www.nceo.ac.uk/data-facilities/datasets-tools/?dataset\_type=tools">https://www.nceo.ac.uk/data-facilities/datasets-tools/?dataset\_type=tools</a>, NCEO, last access: 5 May 2025).

# **Data Availability**

The L2 column carbon dioxide data from OCO-2 and OCO-3 are available from the Goddard Earth Sciences Data and Information Services Centre (<a href="https://doi.org/10.5067/E4E140XDMPO2">https://doi.org/10.5067/E4E140XDMPO2</a>; last access 5 May 2025). The GOSIF GPP is available for public from <a href="https://data.globalecology.unh.edu/data/GOSIF-GPP\_v2">https://data.globalecology.unh.edu/data/GOSIF-GPP\_v2</a> (last access 5 May 2025). The MODIS EVI of version v06.1 is available from <a href="https://lpdaac.usgs.gov/products/myd13a3v061/">https://lpdaac.usgs.gov/products/myd13a3v061/</a> (last access 5 May 2025).

#### **Author contributions**

LF and PIP designed the research with contributions from LS; LF prepared the calculations; PIP and LF wrote the paper; JX, PC, AL, OH, RK, SM, SMP, XR, and MS provided data and comments on the manuscript.

# **Competing interests**

None of the authors has any competing interests.

#### Acknowledgements

We gratefully acknowledge science teams of the NASA Orbiting Carbon Observatory. We also thank the GEOS-Chem community, particularly the team at Harvard University who helped to maintain the GEOS-Chem model, and the NASA Global Modeling and Assimilation Office (GMAO) who provided the MERRA2 data product. PIP would like to acknowledge Chris O'Dell and Tommy Taylor (CSU) for their insights into OCO-2 data processing.

#### **Financial support**

PIP and LF received support from the UK National Centre for Earth Observation funded by the Natural Environment Research Council (grant no. NE/R016518/1) and from the UK Space Agency. JX. was supported by the National Science Foundation (Macrosystem Biology & NEON-Enabled Science program: DEB-2017870) and the Iola Hubbard Climate Change Endowment. XR is funded by The National Institute of Standards and Technology, USA (#70NANB18H16). ICOS activities

at CMN are supported by the JRU-ICOS Italy and by Project ITINERIS – Italian Integrated Environmental Research Infrastructures System (Project code IR0000032) within PIANO NAZIONALE DI RIPRESA E RESILIENZA, MISSIONE 4, COMPONENTE 2, INVESTIMENTO 3.1 "Fondo per la realizzazione di un sistema integrato di infrastrutture di ricerca e innovazione", which also supports the Simonetta Montaguti's position. Measurements at Jungfraujoch were supported by ICOS Switzerland (SNF grant 20F120 198227).

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

# **Figures**

Figure 1. Atmospheric growth rates of CO<sub>2</sub> (blue) and their annual change (black). a Global mean values. b Values determined from Mauna Loa, Hawaii CO<sub>2</sub> mole fraction data. Data collected by NOAA and available at <a href="https://gml.noaa.gov/ccgg/trends/gl\_gr.html">https://gml.noaa.gov/ccgg/trends/gl\_gr.html</a>. c Multi-decadal changes in the probability density of global mean annual mean growth rates and d as panel c but using data from Mauna Loa. Blue and black horizontal dashed lines denote the 1-σ and 2-σ values for the annual atmospheric CO<sub>2</sub> growth and its annual change, respectively.

**Figure 2.** Annual mean *a posteriori* CO<sub>2</sub> flux estimates inferred from OCO-2 data for the globe, the southern extratropics, the tropics, and the northern extratropics. The thin black vertical lines denote the 1-sigma values about the annual mean values. The red lines in panels b-d denote the percentage contribution to the global net fluxes.

**Figure 3.** Differences in *a posteriori* CO<sub>2</sub> flux estimates inferred from OCO-2 data (top), gross primary production (GPP) estimated from OCO-2 SIF data (middle), and elevated vegetation indices (EVI) inferred from MODIS data (bottom) for 2022-2023 (left panels) and 2014-2015 (right panels). Rectangles shown in panel a describe the geographical regions we focus on for our multivariate fits.

**Figure 4.** Differences in surface temperature (Temp; top row), precipitation (Prec; second row), soil moisture (SM; third row), vapour pressure deficit (VPD; fourth row), derived from soil moisture, based on MERRA2 reanalyses data products from NASA GSFC GMAO, and liquid water equivalent (LWE; bottom row) from the GRACE satellites for 2023 minus 2022 (left panels) and 2015 minus 2014 (right panels).

**Figure 5.** Regional linear (black) and quadratic (blue) multivariate fits of NBE anomalies (red) inferred from OCO-2 data using independent estimates of rainfall, surface temperature, and soil moisture from MERRA reanalyses data products from NASA GSFC GMAO. Regional definitions, defined in panel a of Figure 3, include East Amazon, tropical East Africa, southern Africa, tropical Asian, and Southeast Asia. Numbers shown inset of each panel include the Pearson correlation coefficient for each fit, and the p-value that corresponds to both fits.

**Figure A1. a** The distribution of 488 sub-regions – including 356 land regions and 132 oceanic regions – for which we report monthly a posteriori CO2 flux estimates inferred from OCO-2 data. **b** The geographical locations of the ground-based measurements of CO2 mole fraction.

**Figure A2.** As Figure 5, but for NBE anomalies inferred using OCO-2 land nadir, land glint, and ocean glint data, and *in situ* data (LNLGOGIS).

**Figure A3.** As Figure 2 but for *a posteriori* CO<sub>2</sub> flux estimates across the tropics. Regions are as defined by the rectangles shown in Figure 3a. Percentage values higher than 100% are a consequence of some regional fluxes being negative.

Figure A4. As Figure 5 but fitting to GOSIF GPP anomalies.

|               | E. Amazon | NAf   | SAf   | Tr.Asia | SE.Asia |
|---------------|-----------|-------|-------|---------|---------|
| Rain          | -0.05     | -0.21 | 0.18  | 0.11    | -0.42   |
| Surface       | 0.40      | 0.09  | 0.17  | 0.06    | -0.03   |
| temperature   |           |       |       |         |         |
| Soil moisture | -0.29     | -0.51 | -0.44 | -0.84   | -0.11   |

**Table A1.** Normalized linear fitting coefficients for the independent variables of the MERRA2 rain, surface temperature, and soil moisture used to fit the NBE anomalies (Figure 5) for the regions defined in Figure 3a, 2014—2023, inclusively. The largest coefficient for each region is highlighted.

| Region   | Rain   | Temp | Soil Moisture | VPD  | GOSIF GPP |
|----------|--------|------|---------------|------|-----------|
| E.Amazon | < 0.01 | 0.34 | 0.31          | 0.01 | 0.02      |
| NAf      | < 0.01 | 0.06 | 0.51          | 0.24 | 0.03      |
| SAf      | < 0.01 | 0.06 | 0.13          | 0.05 | 0.66      |
| Tr. Asia | < 0.01 | 0.01 | 0.32          | 0.01 | 0.44      |
| SE.Asia  | < 0.01 | 0.02 | 0.07          | 0.15 | 0.62      |

**Table A2.** Permutation importance of MERRA2 rain, surface temperature, and soil moisture, VPD, and GOSIF GPP to fit the NBE anomalies (Figure 5), 2014—2023, inclusively, for the regions defined in Figure 3a. The largest contributor for each region is highlighted.

# Appendix B

## **Sensitivity experiments**

To test the robustness of our results, we report the results from other calculations in which we alter one aspect of the inversion.

655 The experiments are described in Table B1. Text in bold denotes the change from our control run (CTRL).

| Experiment | Wind fields | Observation                                                                                                       | Prior flux                                                                                                                                            |
|------------|-------------|-------------------------------------------------------------------------------------------------------------------|-------------------------------------------------------------------------------------------------------------------------------------------------------|
| CTRL       | MERRA2      | Surface CO2 data<br>(113 sites of the Obspack data<br>collection)<br>OCO-2 XCO2 data over land.                   | Monthly ODIAC Fossil Fuel Emissions Monthly Takahashi Ocean flux climatology (scaled) 3-hourly CASA Biospheric flux Monthly fire emission (GFED v4.0) |
| GEOSFP     | GEOSFP      | Surface CO2 data<br>(113 sites of the Obspack data<br>collection)<br>OCO-2 XCO2 data over land.                   | Monthly ODIAC Fossil Fuel Emissions Monthly Takahashi Ocean flux climatology (scaled) 3-hourly CASA Biospheric flux Monthly fire emission (GFED v4.0) |
| LNLGOGIS   | MERRA2      | Surface CO2 data (113 sites of the Obspack data collection) OCO-2 XCO2 data over land. OCO-2 XCO2 data over ocean | Monthly ODIAC Fossil Fuel Emissions Monthly Takahashi Ocean flux climatology (scaled) 3-hourly CASA Biospheric flux Monthly fire emission (GFED v4.0) |
| SIB3-JENA  | MERRA2      | Surface CO2 data (113 sites of the Obspack data collection) OCO-2 Land data. OCO-2 XCO2 data over ocean           | Monthly ODIAC Fossil Fuel Emissions  Monthly Jena Ocean flux climatology  3-hourly SiB3 Biospheric flux  Monthly fire emission (GFED v4.0)            |

**Table B1.** Configurations of our control run and three sensitivity experiments. Underlined text denotes the change from our control run (CTRL)

The GEOSFP inversion is driven by GMAO Goddard Earth Observing System Forward Processing (GEOS-FP) meteorological analyses, based on a convection scheme that is different from the one used in MERRA2 reanalysis, which we use in our control experiment (CNTRL). For the inversion using OCO-2 land nadir, land glint, and ocean glint data, and *in situ* data (LNLGOGIS) inversion, we use additional OCO-2 XCO2 sun-glint retrievals collected over the oceans. The SIB3-JENA inversion includes alternative *a priori* estimates for sea–air CO<sub>2</sub> fluxes based on CO<sub>2</sub> observations (Rödenbeck et al., 2022) and for biosphere-atmosphere fluxes from the SiB3 model simulation (Baker et al., 2008).

Figure B1 compares the monthly *a posteriori* net CO<sub>2</sub> flux estimates, 2014-2024, from our control and the three sensitivity experiments over four TransCom-3 regions, representative of three different latitude ranges: tropical South America, tropical Asia, temperate Eurasia, and South Africa. The a posteriori estimates are very similar, but we find significant regional differences for some months. For example, GEOSFP results in smaller emissions from Temperate Eurasia during winter months (Fig. B1c) and including OCO-2 oceanic glint data results in larger seasonal cycles over Tropical South America (Fig. B2d). As a result, the two inversions that use the ocean data (LNLGOGIS and SIB3-Jena) show net annual emissions from Tropical South America that are 0.1-0.22 PgC yr<sup>-1</sup> lower than the control run.

680

685

670

**Figure B1**. Monthly regional flux estimates by four inversion experiments (CNTRL, GEOSFP, LNLGOGIS and SIB3-JENA) over four TransCom-3 regions: a) Tr. SAm (Tropical South America), b) Tr.As (Tropical Asia), c) TEr (temperate Eurasia), and d) Saf (South Africa). The uncertainties for *a priori* and *a posteriori* estimates from the inversions are denoted by vertical lines, and shaded envelopes, respectively.

Figure B2 shows that the corresponding year to year changes in the natural flux changes between 2022 and 2023, associated with our main conclusion, are remarkably similar over almost every TranCom-3 land region. The ocean estimates appear to depend on using the ocean glint measurements. The two inversions that assimilate only OCO-2 land data (CNTRL and GEOSFP) absorbed 0.4-0.45 PgC yr<sup>-1</sup> less carbon between 2022 and 2023 while the two inversions that also use the sun-glint

measurements (LNLGOGIS and SIB3-JENA), and use a different ocean *a priori* show little change in the ocean net flux between the two years.

695

700

**Figure B2.** Changes in *a posteriori* net biosphere exchange flux estimates (2023 minus 2022) over TransCom-3 regions, estimated by four experiments (Table B1). Vertical lines denote *a posteriori* uncertainties.

Figure B3 compares the correlations between regional CO<sub>2</sub> NBE flux anomalies and anomalies in environment variables between 2014 to 2023. The NBE flux anomalies for tropical South America for our control and the three sensitivity calculations (Table B1) show strong correlations (> 0.5 and a p value 

Figure B3. Pearson correlation coefficients, r, between regional *a posteriori* estimates of net biosphere CO<sub>2</sub> exchange anomalies and anomalies of environmental variables, including (a) MODIS EVI, (b) GOSIF GPP, (c) MERRA2 soil moisture, (d) MERRA2 surface temperature, (e) MERRA2 VPD, and (f) MERRA2 precipitation. Correlations with p value > 0.1 (less significant) are denoted by black hatching line.

|               | E. Amazon (range) | NAf<br>(range) | SAf<br>(range) | Tr.Asia<br>(range) | SE.Asia<br>(range) |
|---------------|-------------------|----------------|----------------|--------------------|--------------------|
| Rain          | -0.05             | -0.21          | 0.18           | 0.11               | -0.42              |
|               | (-0.06, -0.01)    | (-0.32, -0.17) | (-0.03, 0.18)  | (-0.01, 0.30)      | (-0.42, -0.36)     |
| Surface       | 0.40              | 0.09           | 0.17           | 0.06               | -0.03              |
| temperature   | (0.38, 0.48)      | (0.01, 0.24)   | (0.17, 0.32)   | (-0.06, 0.06)      | (-0.03, 0.25)      |
| Soil moisture | -0.29             | -0.51          | -0.44          | -0.84              | -0.11              |
|               | (-0.46, -0.29)    | (-0.56 -0.23)  | (-0.47, -0.40) | (-0.86, -0.78)     | (-0.11,0.27)       |

**Table B2.** As Table A1 but with values reported as a range from the control and the three sensitivity inversions.

# Appendix C

# A posteriori net biosphere CO2 flux estimates for 2024

We extend our control inversion experiment to the end of 2024. Figure C1 shows the difference of *a posteriori* NBE CO<sub>2</sub> flux estimates between 2024 and our baseline year 2022 alongside the difference between 2023 and 2022. Figure C2 shows the same data but broken down into TransCom-3 regions. We find that tropical land absorbed less carbon in 2024 than during 2022, primarily over South America, Africa, and to a lesser extent Southeast Asia.

Our calculations correspond to a net global annual CO<sub>2</sub> emission of 6.84±0.80 PgC yr<sup>-1</sup>, equivalent to global CO<sub>2</sub> growth rate of 3.28±0.30 ppm for 2024. During 2023 and 2025, we estimate from OCO-2 data that atmospheric levels of CO<sub>2</sub> increased by 6.36 (3.09+3.28) ppm compared to 6.48 (2.76+3.72) ppm inferred from the NOAA surface network.

**Figure C1.** Changes in annual mean *a posteriori* NBE flux estimates from our control inversion between (a) 2022 and 2023 and between (b) 2022 and 2024.

**Figure C2**. Changes in annual mean *a posteriori* NBE flux estimates from our control inversion between 2022 and 2023 and between 2022 and 2024 for TransCom-3 regions. Vertical lines denote *a posteriori* uncertainties.