# Peer review of "The role of the tropical carbon balance in determining the large atmospheric CO2 growth rate in 2023"

_EGUsphere, 2025_

## Author Comment (AC1)

Below we provide point by point responses to the comments provided by Hartmut Boesch and the two anonymous reviewers. The original comments are in italics.

**Comment from Hartmut Boesch**

*The study of Feng et al. focusses on the interpretation of CO2 fluxes during the year 2023 with a record high CO2 growth. The papers shows that reduced CO2 uptake by land ecosystems, eg Brasil, likely due to warmer temperatures is the primary driver for the high CO2 growth. The year 2023 has been followed by another year of with a similarly high CO2 growth rate in 2024. The year 2023 was characterised by moderate positive values of the El Nino index (indicative of an El Nino) in the second half of the year which continued for a few months into 2024 and which turned into negative values in the second half of the year. Thus, contrasting the year 2024 against 2023 would be of great interest and would help to put the findings for the year 2023 into a broader context. Since data from the NASA OCO-2 satellite for the year 2024 is readily available, I was wondering if the study could be extended to include another year.*

**This is great suggestion. We have now included the results from 2024 in Appendix C, which provide a more complete analysis of the El Niño. We have introduced the 2024 analysis in the concluding remarks section to avoid diverting the reader from the results reported in the main study.**

**We find that the reduced carbon uptake continues into 2024. Uptake by the Amazon basin in 2024 remains weaker than in 2022. There is also weakened uptake over southern tropical Africa (south of 20ºS) and over tropical Asia. There is a small increase in uptake over temperate North America in 2024 compared to 2023. The resulting global net emission estimate for 2024 is 6.84±0.80 PgC, corresponding to a global CO2 growth rate of 3.28±0.30 ppm/yr.**

**Reviewer 1**

*General comments.*

*Authors present a top-down look at global/regional carbon cycle response to initial (year 2023) stage of 2023/2024 El Nino event. In the anomaly analysis, they look at the period covered by OCO-2 data from 2015 to 2023. To go beyond the bare flux estimates by the inverse model they use the data by remote sensing of vegetation productivity and hydrology to analyze regional scale variations of net biosphere exchange, suggesting the largest contribution to net $CO_2$ emissions was from tropical South America, with a number of other regions responding with increased uptake or emissions. They also found some regional discrepancies with another study by Ke at al. 2024 which is based on another inverse model and points to a need to reconcile the differences in the future. The paper is well written and can be accepted after minor revisions.*

*Detailed comments:*

*To complement the tropical mean flux figures, would be very useful to note a TRENDY analysis by Sitch et al (2024), in which the contributions of $CO_2$ fertilization and climate change to the regional NEE trends were evaluated separately and a detrimental impact of warming on many tropical ecosystems was shown.*

**Excellent suggestion. We have added that reference.**

*Line 150. Notably, Pandey et al (2024) estimated the extent to which the NOAA surface network-based growth rate can depart from "true" one based on whole-atmosphere total $CO_2$ annual change.*

**That's a great point. We have now added that reference.**

*References:*

*Pandey, S., Miller, J. B., Basu, S., Liu, J., Weir, B., Byrne, B., et al. (2024). Toward low-latency estimation of atmospheric CO2 growth rates using satellite observations: Evaluating sampling errors of satellite and in situ observing approaches. AGU Advances, 5, e2023AV001145, https://doi.org/10.1029/2023AV001145*

*Sitch, S., O'Sullivan, M., Robertson, E., Friedlingstein, P., Albergel, C., Anthoni, P., et al. (2024). Trends and drivers of terrestrial sources and sinks of carbon dioxide: An overview of the TRENDY project. Global Biogeochemical Cycles, 38, e2024GB008102, https://doi.org/10.1029/2024GB008102*

**Reviewer 2**

*The study by Feng et al. aims to attribute the unusually large atmospheric $CO_2$ growth rate in 2023 to regional drivers. Using an atmospheric inversion based on $CO_2$ concentrations from the OCO-2 satellite and ground-based in-situ networks, the authors derive net biosphere exchange (NBE) fluxes and identify the tropics as the dominant contributing region. Through a basic correlation analysis with environmental variables, they suggest that elevated temperatures in Brazil and moist conditions in other tropical regions may explain the observed tropical NBE anomalies.*

*However, I find the study lacks robustness and a transparent discussion of potential uncertainties. Several conclusions appear to be based on apparent tendencies in the data rather than physically robust patterns. In places, the narrative appears to drive the interpretation of results, rather than the analysis guiding the narrative. Consequently, I do*

*not recommend publication of the manuscript in Atmospheric Chemistry and Physics. Below, I outline the major concerns:*

**We disagree with this assessment of our study. Below, we have provided responses to these comments.**

*Robustness of the atmospheric inversion:*

*The study presents NBE estimates for 2014–2023 but focuses primarily on 2022–2023 (and, to some extent, 2014–2015) to explain the 2023 $CO_2$ growth rate. These conclusions are drawn solely from a single inversion system (GEOS-Chem/MERRA-2), with no accompanying discussion on the reliability or limitations of this system and the uncertainties involved.*

**This is an established system that is routinely evaluated using all available data. The model also plays a role in the annual Global Carbon Project where it is compared against other model results. It was also part of the NOAA MIP project (see https://gml.noaa.gov/ccgg/OCO2_v7mip/, and Crowell, et al, 2019), Our posterior model concentrations have also been used in evaluation and bias-correction of OCO-2 retrievals (see for example, O'Dell et al., 2019).**

**But we agree with the reviewer about uncertainties, particularly those associated with errors in transport model and the inversion configuration, e.g. assumed prior fluxes and the selection of observations. To address this point we have added the results from three sensitivity experiments in Appendix B to demonstrate the model performance.**

*For earlier years, the OCO-2 Model Intercomparison Project (MIP) provides an ensemble of atmospheric inversions. To build confidence in their analysis for 2023, I strongly recommend the authors compare their NBE estimates with those from the OCO-2 MIP ensemble for overlapping years. Incorporating error estimates—such as the spread across MIP ensemble members—would allow for a better assessment of the impact of model choices on the results. A comparison of a posteriori regional flux distributions would also be essential to evaluate whether different inversion systems provide consistent spatial signals, especially given the sparse observational constraints in tropical regions. The tropical signal remains difficult to constrain even with OCO-2 data due to persistent cloud cover and the South Atlantic magnetic anomaly, which degrades measurement quality over South America. Further strengthening of the study could be achieved by including inversions using GOSAT $XCO_2$ data, which are available from 2009 onward.*

**Assessing the performance of a model cannot be achieved conclusively by comparing against other models, particularly when there remain large, unexplained**

discrepancies. That approach provides information about the spread in models, which is useful when, for example, reporting regional or countryside carbon budgets. While our estimates are within the range shown by other models reported in the ongoing OCO-2 MIP v11 project (not shown), our focus has been to test our posterior fluxes with independent data for which our model consistently performs well (Taylor et al, 2023).

The result we present with our model configuration is our solution that we can tie back to independent data. It reproduces the reported global atmospheric growth rate of CO2 and it is consistent with the temporal and spatial variations of related quantities such as solar induced fluorescence and hydrological data. To address this reviewer's point, we have clarified the meaning of our posterior solution and emphasized that we are more interested in different IAV during El Niño or La Niña, which as shown in Appendix B is more reliable under different inversion configurations. While we agree the inability to estimate fluxes over many tropical lands such, as the Tropical America, at high spatial resolution, available OCO-2 data can still provide insights on the Tropical South America as whole, as indicated by large departure from the prior estimates, and by the significantly reduced uncertainty shown in (new) Figure B1a.

We chose to use OCO-2 for this study because the assimilated dataset was consistently generated using an algorithm that has been extensively evaluated. It provides stable coverage till present (Das et al., 2025). The GOSAT sensor and, to a lesser extent, the instrument pointing system have experienced changes in recent years (Someya et al, 2023), which will impact the resulting CO2 flux distributions. Considering the systematic bias in available GOSAT and OCO-2 retrievals, a more reliable approach may be to use a merged GOSAT and OCO-2 XCO2 dataset such as the one currently produced by the University of Bremen. Use of the merged data will be the subject of future work.

We agree that cloud cover (wet season) and biomass burning aerosol (dry season) limit clear-air measurements across key tropical continental region. However, tropical CO2 fluxes are typically inferred from data collected downwind of the continent, a result that was highlighted before OCO-2 was launched (see Palmer et al., 2011).

Regarding the South Atlantic Anomaly (SAA), there is little evidence that OCO-2 suffers from this phenomenon. This is because of the way the OCO-2 retrieval team screen cosmic rays at the pixel level from individual soundings. We refer the reader to Crisp et al, (2017) who provide a thoughtful discussion on the topic. Taylor et al (2023) show the distribution of "good" OCO-2 sounding. There are fewer soundings

**over South America because of the SAA but cloud represent the bigger observational challenge.**

*If the authors believe their inversion setup is particularly well-suited to the scientific question, they should provide supporting evidence and/or mechanistic justification for its advantages over other inversion setups. This could be achieved by collecting sensitivity studies testing the robustness of the regional flux distributions and by examining the match of posterior simulations to the measurement data.*

**We are not sure how to respond to this reviewer comment. We are one of several competing groups around the world who have this capability. The result reported in our study builds on decades of model development and evaluation. We cannot comment on other group inversion setups. But there is no reason to suggest that our model is not well-suited to address the science being reported.**

**As discussed above in response to another comment from this reviewer, we have now reported in Appendix B a series of sensitivity tests to examine the robustness of our result. Based on those tests, we have confidence that the result we have reported is robust. These results support our main conclusion that the high CO2 growth rate in 2023 is substantially influenced by reduced uptake over tropical land regions, particularly over tropical South America and Tropical Asia.**

**Also mentioned, above our model participates regularly with international model intercomparisons (e.g., Figure S5, Friedlingstein et al (2024)). And as part of our own internal model evaluation, we compare our model against a range of data. A summary evaluation of TCCON data (2014-2023) reveals that the model bias is generally smaller than 0.5 ppm and the standard deviation is smaller than 1.2 ppm, consistent with our previous studies. These values are consistent with the OCO-2 MIP v9 project highlighted by this reviewer – see Figures 6 and 7 from Byrne et al, (2023). We also find that our posterior CO2 fluxes from OCO-2 reproduce the trend and seasonal cycle of the TCCON XCO2 data at different latitudes, typically within 0.2 ppm.**

Attribution to environmental drivers:

*The study attempts to attribute interannual differences in NBE (e.g., between 2022 and 2023) to environmental anomalies by fitting linear or quadratic functions of temperature and moisture variables to the flux anomalies. This approach, as described (albeit vaguely), essentially bypasses the complexity of biospheric carbon exchange processes. I recommend the authors instead compare their inversion-derived NBE anomalies with process-based global vegetation models—such as those from the TRENDY ensemble— which account for mechanistic responses of terrestrial ecosystems to environmental*

*drivers. Even if vegetation models are unavailable for 2023, such comparisons for previous years would help contextualize the findings and build trust in the methodology.*

**There appears to be a fundamental divide in our modelling philosophies. All models are wrong, but some are useful. Evaluating our posterior fluxes with models is, in our opinion, less useful than testing their consistency with independent data. What we have done in this study is to understand to what extent changes in temperature and water – key quantities on the scales observed by OCO-2 and modelled – can explain changes in the posterior fluxes inferred from OCO-2.**

**We have not claimed anything more than that. Reconciliation with process-based global vegetation models is well beyond of the scope of this study, partially due to poor observation coverage and uncertainty for both $CO_2$ data, and other land/atmospheric properties.**

**However, our results, now including Figure B3, highlight the possibility of a model intercomparison at large spatial scales. We have chosen years when there are significant changes in $CO_2$ fluxes that can exaggerate deficiencies in top-down and bottom-up flux estimates.**

*Section I.176–191 (regional NBE patterns):*

*This section discusses regional NBE, GPP, and EVI differences for select years, primarily through visual inspection of maps in Fig. 3. The discussion is qualitative and, in parts, unconvincing. For example, the claim that "a posteriori fluxes are broadly consistent with independent estimates of GPP" (l.186) is difficult to substantiate visually, particularly in the tropics, where correlations between GPP and NBE anomalies appear weak. This section would benefit from a more quantitative assessment of how regional patterns contribute to net flux changes, and should include uncertainty estimates to distinguish statistically significant signals. The choice of focus regions in Fig. 3/5 also appears arbitrary and should be justified—ideally, based on inversion information content and ecological boundaries, rather than visual inspection.*

**The coarse spatial resolution of the posterior $CO_2$ fluxes, and the uncertainties in the environmental datasets preclude a detailed examination of their correlation at fine spatial scales. Here, we have examined the correlations at spatial scales commensurate with the posterior $CO_2$ fluxes, which will be diluted by heterogenous biosphere response. Nevertheless, we reveal some interesting correlations at the subcontinental scale (Figure B3). As part of our updated analysis, we have reported correlations over five subregions across the tropics, based on the large changes in posterior $CO_2$ fluxes we report between 2014 and 2015 and between 2022 and 2023. We show in Figures 5 and B3 the analysis for**

**different sub-regions in tropics and report the corresponding statistics in Table B2. We have also added text that described the linear (f1) and quadratic (f2) models used to fit posterior NBE anomalies and clarified the contents of Tables A1 and A2.**

*Section I.201–214 and Fig. 5 (parameter fits):*

*The discussion of parameter fits to flux anomalies is overly optimistic. For example, in Fig. 5, the fits reproduce the strong anomalies in 2015 and 2023 only approximately and fail to capture much of the variability in other years. No error estimates are provided. Furthermore, the methodology for fitting environmental parameters is insufficiently described. The functions $f_1$ and $f_2$ in Fig. 5 are undefined, and the meanings of Tables A1 and A2 are unclear. This section requires a clearer and more rigorous presentation of methods and uncertainties to support its conclusions.*

**The coarse spatial resolution of the posterior CO2 fluxes, the uncertainties in the environmental datasets, and the non-trivial responsible physical processes limits our ability to explain perfectly CO2 flux anomalies with environmental data. Instead, our focus is on understanding which parameters are the most important for individual regions (Figure 5 and Table B2). As discussed above, we also examine the results from three different inversion configurations to explore the robustness of our results (Table B2).**

*Section I.251-261 (conclusions)*

*The discussion on findings of other studies is too short, in particular since other studies draw different conclusions despite using similar measurements. A discussion needs to include the role of transport model errors, assumptions on prior constraints and information content in the tropics.*

**There are many differences between flux inversions. These include the inversion approach (e.g., 4D-Var vs ensemble Kalman filter), atmospheric transport models (e.g., TM5 vs GEOS-Chem), as well as choices about prior fluxes, etc. We have now included an analysis on this and explored different assumptions.**

**Assessing the performance of a model cannot be done by comparing against other models. That approach provides information about the spread in models. Our focus has been to test our posterior fluxes with independent data for which our model consistently performs well (Taylor et al, 2020). Based on those tests, we have confidence that the result we have reported is robust. These results support our main conclusion that the high CO2 growth rate in 2023 is substantially influenced by reduced uptake over tropical land regions, particularly over tropical South America and Tropical Asia.**

*Given the methodological concerns outlined above, the statement speculating about "the beginning of a decline in the ability of tropical ecosystems to absorb carbon" (l.263) is premature and risks being perceived as sensationalist.*

**What our concluding remarks do say is:**

**"If our main result is accurate – a moderate El Niño event has led to a significant reduction in carbon uptake by the tropical land biosphere, which has experienced extensive drought – we might be observing the beginning of a decline in the ability of tropical ecosystems to absorb carbon."**

**The words are chosen carefully to avoid sensationalism.**

*The conclusion section should be revised to more accurately reflect the limited scope of the analysis and the significant uncertainties involved. The current framing exceeds what the data and methodology can confidently support. A similar concern applies to the abstract, where the statement, "Our results suggest that ongoing environmental degradation of the Amazon is now playing a substantial role in increasing the global atmospheric $CO_2$ growth rate" (line 34), is not adequately supported by the study's findings.*

**We disagree with this statement. Our conclusion provides a balanced view of what could be an unfolding situation. Note the next statement after the one that mentions the potential decline in the ability of tropical ecosystems to absorb carbon is this:**

**"The long-term nature of this situation is unclear without further data, although the 265 preliminary estimate of the 2024 atmospheric CO2 growth rate of 3.75±0.08 ppm/yr is unprecedented since these records began in the late 1950s (https://gml.noaa.gov/ccgg/trends/gl_gr.html; last access: 15th April 2025). A coordinated measurement campaign is urgently needed to document how tropical ecosystems are changing, whether these changes compromise the future ability to absorb and store carbon, and whether prolonged drought will substantially delay any ecosystem recovery."**

**In other words, we are couching our concluding remarks with substantial caveats but also acknowledge we need to get much better information on the ground to understand what is going on. This is urgent because we have no idea whether this unprecedented situation will continue.**

Regarding the abstract, our result *do* support this statement. The Amazon basin has suffered from extensive and widespread drought (environmental degradation) that has now weakened its ability to absorb carbon to such an extent that it has influenced the global growth rate of atmospheric CO2. Again, we have been careful with our words and "suggest that" that could be substituted for "are consistent with."

**References**

Byrne, B., et al: National $CO_2$ budgets (2015–2020) inferred from atmospheric $CO_2$ observations in support of the global stocktake, Earth Syst. Sci. Data, 15, 963–1004, https://doi.org/10.5194/essd-15-963-2023, 2023.

Crisp, D., et al: The on-orbit performance of the Orbiting Carbon Observatory-2 (OCO-2) instrument and its radiometrically calibrated products, Atmos. Meas. Tech., 10, 59–81, https://doi.org/10.5194/amt-10-59-2017, 2017.

Crowell S., D. Baker, A. Schuh, S. Basu, A. R. Jacobson, F. Chevallier, J. Liu, F. Deng, L. Feng, K. McKain, et al. The 2015–2016 carbon cycle as seen from oco-2 and the global in situ network. Atmospheric Chemistry and Physics, 19(15):9797–9831, 2019.

Das, S., et al. (2025). Comparisons of the v11.1 Orbiting Carbon Observatory-2 (OCO-2) XCO2 measurements with GGG2020 TCCON. *Earth and Space Science*, *12*, e2024EA003935. https://doi.org/10.1029/2024EA003935

Friedlingstein, P., et al: Global Carbon Budget 2024, Earth Syst. Sci. Data, 17, 965–1039, https://doi.org/10.5194/essd-17-965-2025, 2025.

O'Dell, C. W., et al: Improved retrievals of carbon dioxide from Orbiting Carbon Observatory-2 with the version 8 ACOS algorithm, Atmos. Meas. Tech., 11, 6539–6576, https://doi.org/10.5194/amt-11-6539-2018, 2018.

Palmer, P. I., Feng, L., and Bösch, H.: Spatial resolution of tropical terrestrial CO2 fluxes inferred using space-borne column CO2 sampled in different earth orbits: the role of spatial error correlations, Atmos. Meas. Tech., 4, 1995–2006, https://doi.org/10.5194/amt-4-1995-2011, 2011.

Someya, Y., et al: Update on the GOSAT TANSO–FTS SWIR Level 2 retrieval algorithm, Atmos. Meas. Tech., 16, 1477–1501, https://doi.org/10.5194/amt-16-1477-2023, 2023.

Taylor, T. E., et al.: Evaluating the consistency between OCO-2 and OCO-3 $XCO_2$ estimates derived from the NASA ACOS version 10 retrieval algorithm, Atmos. Meas. Tech., 16, 3173–3209, https://doi.org/10.5194/amt-16-3173-2023, 2023.

---

## Editor Decision (ED1)

**Editor comments egusphere-2025-1793 Revised**

There are a lot of small issues that should be corrected before the manuscript be accepted for publication. See my detailed list below.

**General comments:**

One major point of criticism I have is that I find it quite difficult to follow what actually has be done and how. Since you have several papers published using the same method important details are omitted. Just writing this is a well established method and then listing a bunch of papers is not enough. You cannot expect every reader or referee to read 4 or 5 additional papers. I had quick look through them and found that the Feng et al. papers provide some more information on the method.

I would appreciate if the most important points about the used method could be repeated in this manuscript with clearly referencing the corresponding papers at the respective places so that the reader knows where the more detailed information is provided.

What also became not clear to me (being a non-expert on this topic) is if all figures show model data or are in some figures also pure measurement data shown? Has one model experiment been used and shown or have here several different model runs been used and shown.

I also had trouble understanding how you came to your conclusion based on your results. Also here, I have the feeling that a lot of important information is missing. So please have a careful check through the manuscript and add what is missing.

**Specific comments:**

- P1, L25: Add OCO-2 in parentheses after Orbiting Carbon Observatory
- P3, L82 and throughout the manuscript: Section should be abbreviated as "Sect." unless it appears at the begin of the sentence.
- P4, L102: Time period somewhat lost here. Is that the time period of the measurements considered? What is the time period for which the model run has been made? Has here only one model run been made/considered for this study or several?
- P4, L110: Abbreviation JPL-ACOS should be introduced.
- P4, L120: Abbreviations of well-known institutions like NOAA (and also in the abstract NASA) may be introduced as well.
- P4, L122: Either add the abbreviation "GPP" in parenthesis or write "gross primary productivity", using small letters as first letters.
- P4, L124: Add "at" -> available globally at a spatial......
- P4, L126: Add "of CO2" after "annual mean global total" to be more clear.
- P5, L131: Add "GRACE" in parentheses after "experiment" or write "recovery and climate experiment" (using small letters as first letters).
- P5, L139: Reanalyses -> reanalyses
- P5, L140: Better to use a small s as subscript than writing "TS"?
- P5, L134, L142 and 155: Remove comma after "et. al.".

- P5, L136: Introduce abbreviations.
- P5, L153: ppm/year -> ppm year-1 (write units according to the Copernicus style, see manuscript preparation guidelines)
- P5, L157: Add "TCCON" in parentheses after "Network"
- P6, L175: 2015/2015? You mean 2015/2016?
- P6, L183: remove comma after et a.
- P6, L190: Abbreviation "SIF" should be introduced.
- P7, L195: Add "the" and "of" so that it reads "describe the large-scale changes of CO2 fluxes"
- P7, L198: Don't start a new sentence with "And". Please rephrase.
- P7, L201: dynamic -> dynamical?
- P7, L207: Before the surface temperature was introduced as "TS", not it is only "T". Use a consistent naming throughout the manuscript.
- P7, L225: What is meant here with the time period? In this sentence this does not make sense. I would suggest to mention the time period considered already at the begin of the paragraph where Figure 5 is described.
- P8, L226: Concluding Remarks -> Conclusion

Note, the conclusion is too long. I would suggest to split this section and to have a discussion or summary section and a short conclusion section where the major findings and implications are summarized.

- P8, L231: Further examination of what? Please be more clear.
- P8, L231: "foci" -> please rephrase.
- P8, L240: Could you please refer to the respective figure or section of the manuscript?
- P8, L251: change -> change of
- P8, L254: In 2023 appears twice in the sentence, one is obsolete and should be removed.
- P9, L261: remove comma after e.g.
- P9, L266: over temperate North America not clear, please rephrase. Why "temperate"? Why not just over North America?
- P9, L267: ppm/yr -> ppm yr-1. Note, before your wrote year, now it is yr. Please use a consistent writing of units and check the ACP guidelines if it should be yr, y, or year.
- P9, L279-280: I have difficulties to follow you here. Why does a moderate El Nino cause extensive droughts? Why does it have now a severe effect, But not in earlier years?
- P11, Reference of Baker et al.: The half of author names is written in capital letters. Please correct to normal writing (names starting with capital letters and the rest with small letters).

Figures: Add a full stop after the figure number.

P21, L577: Rephrase "Number shown inset"

P27, Table A2 caption: The time period is a bit lost year. Write it that way that it is understandable what time period is considered.

P28, Table B1: To my knowledge in printed ACP papers bold text is not allowed. You need to find an other way to emphasize this.

P28, Table 1 caption: Full stop at the end of the sentence is missing.

P28, L613: Also here the abbreviations need to be introduced.

P29, Figure B1: Replace colon after figure number with a full stop.

P29, L637: PgC/yr -> PgC yr-1

P30, Figure B2: Decrease figure or font size. Note, all figures should be prepared in a similar style.

P30, L544: NBE has already been introduced in the manuscript.

P30, L647: Abbreviations "EVI" and "GOSIF" should be introduced.

P31, Figure B3: Replace colon by full stop after figure number and write either Pearson correlation coefficient or put r in parentheses.

P32, Table B2: There is a lost opening parenthesis. Is here the header line missing? Or is that here the continuation of the table from the previous page? Please take care that tables are appear correctly in the manuscript.

P33, L676: Remove text "taking advantage to access to the necessary data during the review process". There is no need to point out what has been done based on the referee comments.

P33, L670: "with loci"? Please rephrase.

P34, Figure C2: Replace colon after Figure number with a full stop and decrease figure or font size (see my comment on this further above).

---

## Author Response (AR2)

**Responses to editorial comments**

Thanks so much for these additional remarks. This study builds on our recent previous studies, so we perhaps relied too much on that material in this study. We have now addressed that point with additional text, as suggested. Except for a few points (highlighted below), we have addressed all the comments. We have tracked changes from the previous (clean) version of the manuscript we uploaded so you can see what we have changed based on your comments. We have also revised Figures 4 and B3 so the labels are consistent with the variable names used in the text.

**General comments:**

One major point of criticism I have is that I find it quite difficult to follow what actually has be done and how. Since you have several papers published using the same method important details are omitted. Just writing this is a well established method and then listing a bunch of papers is not enough. You cannot expect every reader or referee to read 4 or 5 additional papers. I had quick look through them and found that the Feng et al. papers provide some more information on the method. I would appreciate if the most important points about the used method could be repeated in this manuscript with clearly referencing the corresponding papers at the respective places so that the reader knows where the more detailed information is provided.

Fair point. We have now included some text that describes the methodology, including references.

What also became not clear to me (being a non-expert on this topic) is if all figures show model data or are in some figures also pure measurement data shown? Has one model experiment been used and shown or have here several different model runs been used and shown.

The figures show a posteriori CO2 flux estimates inferred from data and the ensemble Kalman filter (EnKF) or data, as described in the captions. All figures in the study use the GEOS-Chem model and the EnKF with the same set of in situ and OCO-2 data. The figures shown in the main part of the paper use the same set of emission inventories and meteorological analyses. In Appendix B we describe sensitivity experiments in which we use different inventories, different meteorology, and different types of data.

I also had trouble understanding how you came to your conclusion based on your results. Also here, I have the feeling that a lot of important information is missing. So please have a careful check through the manuscript and add what is missing.

We have now highlighted the figures needed to follow our argument. There was a missing piece of information. The moderate El Nino event led to an outsized impact on CO2 fluxes over the Amazon basin because this region was already subject to extensive drought that other studies have attributed to changes in climate. This is now clearer and we have included the appropriate reference (Clarke et al, 2024).

**Specific comments:**

Unless otherwise stated, we have addressed all the comments raised. We have used highlighted text to emphasise the points with which we respectively disagree.

P1, L25: Add OCO-2 in parentheses after Orbiting Carbon Observatory

P3, L82 and throughout the manuscript: Section should be abbreviated as "Sect.

" unless it appears at the begin of the sentence.

P4, L102: Time period somewhat lost here. Is that the time period of the measurements considered? What is the time period for which the model run has been made? Has here only one model run been made/considered for this study or several?

P4, L110: Abbreviation JPL-ACOS should be introduced.

P4, L120: Abbreviations of well-known institutions like NOAA (and also in the abstract NASA) may be

introduced as well.

P4, L122: Either add the abbreviation "GPP" in parenthesis or write "gross primary productivity"

using small letters as first letters.

P4, L124: Add "at"

-> available globally at a spatial......

P4, L126: Add "of CO2" after "annual mean global total" to be more clear.

P5, L131: Add "GRACE" in parentheses after "experiment" or write "recovery and climate experiment" (using small letters as first letters).

P5, L139: Reanalyses -> reanalyses

P5, L140: Better to use a small s as subscript than writing "TS"?

P5, L134, L142 and 155: Remove comma after "et. al."

.P5, L136: Introduce abbreviations.

P5, L153: ppm/year -> ppm year-1 (write units according to the Copernicus style, see manuscript

preparation guidelines)

P5, L157: Add "TCCON" in parentheses after "Network"

P6, L175: 2015/2015? You mean 2015/2016?

P6, L183: remove comma after et a.

P6, L190: Abbreviation "SIF" should be introduced.

P7, L195: Add "the" and "of" so that it reads "describe the large-scale changes of CO2 fluxes"

P7, L198: Don't start a new sentence with "And". Please rephrase.

P7, L201: dynamic -> dynamical?

P7, L207: Before the surface temperature was introduced as "TS", not it is only "T". Use a consistent

naming throughout the manuscript.

P7, L225: What is meant here with the time period? In this sentence this does not make sense. I

would suggest to mention the time period considered already at the begin of the paragraph where

Figure 5 is described.

**P8, L226: Concluding Remarks -> Conclusion**

Note, the conclusion is too long. I would suggest to split this section and to have a discussion or summary section and a short conclusion section where the major findings and implications are summarized.

We have used concluding remarks for several of our ACP papers, which merges a succinct conclusion with a summary discussion, including some comments on previous work and the wider implications of our study. We have appreciated this level of editorial

freedom in the past with ACP, which allows the reader to absorb the key points and wider implications of the study in one section.

P8, L231: Further examination of what? Please be more clear.

P8, L231: "foci"

-> please rephrase.

P8, L240: Could you please refer to the respective figure or section of the manuscript?

P8, L251: change -> change of

P8, L254: In 2023 appears twice in the sentence, one is obsolete and should be removed.

P9, L261: remove comma after e.g.

P9, L266: over temperate North America not clear, please rephrase. Why "temperate"? Why not just over North America?

We also have boreal North America (see Figure B2, for example). This nomenclature is defined in the TransCom-3 experiments (for comparing top-down flux inversions) and is used widely in this field to separate it from 'Boreal North America'.

P9, L267: ppm/yr -> ppm yr-1. Note, before your wrote year, now it is yr. Please use a consistent

writing of units and check the ACP guidelines if it should be yr, y, or year.

P9, L279-280: I have difficulties to follow you here. Why does a moderate El Nino cause extensive droughts? Why does it have now a severe effect, But not in earlier years?

P11, Reference of Baker et al.: The half of author names is written in capital letters. Please correct to

normal writing (names starting with capital letters and the rest with small letters).

Figures: Add a full stop after the figure number.P21, L577: Rephrase "Number shown inset"

We weren't sure what you meant this. "Number(s) shown inset of each panel..." makes sense.

P27, Table A2 caption: The time period is a bit lost year. Write it that way that it is understandable

what time period is considered.

P28, Table B1: To my knowledge in printed ACP papers bold text is not allowed. You need to find an other way to emphasize this.

We are sure about this. In any case, we have underlined text. We will explore this with the copyeditors. Our preference would be bold text.

P28, Table 1 caption: Full stop at the end of the sentence is missing.

P28, L613: Also here the abbreviations need to be introduced.

P29, Figure B1: Replace colon after figure number with a full stop.

P29, L637: PgC/yr -> PgC yr-1

P30, Figure B2: Decrease figure or font size. Note, all figures should be prepared in a similar style.

P30, L544: NBE has already been introduced in the manuscript.

P30, L647: Abbreviations "EVI" and "GOSIF" should be introduced.

P31, Figure B3: Replace colon by full stop after figure number and write either Pearson correlation

coefficient or put r in parentheses.

P32, Table B2: There is a lost opening parenthesis. Is here the header line missing? Or is that here

the continuation of the table from the previous page? Please take care that tables are appear

correctly in the manuscript.

P33, L676: Remove text "taking advantage to access to the necessary data during the review

process"

. There is no need to point out what has been done based on the referee comments.

P33, L670: "with loci"? Please rephrase.

P34, Figure C2: Replace colon after Figure number with a full stop and decrease figure or font size

(see my comment on this further above).